# Circulating Tumor Cell-Derived Pre-Clinical Models for Personalized Medicine

**DOI:** 10.3390/cancers11010019

**Published:** 2018-12-24

**Authors:** Marta Tellez-Gabriel, Denis Cochonneau, Marie Cadé, Camille Jubelin, Marie-Françoise Heymann, Dominique Heymann

**Affiliations:** 1RNA and Molecular Pathology Research Group, Department of Medical Biology, The Artic University of Tromsø, N-9037 Tromsø, Norway; 2INSERM, Centre de Recherche en Cancérologie Immunologie Nantes Angers, Team 9 “Apoptosis and Tumor Progression”, Institut de Cancérologie de l’Ouest, Université de Nantes and Université d’Angers, 44805 Saint Herblain, France; denis.cochonneau@ico.unicancer.fr (D.C.); marie.cade@univ-nantes.fr (M.C.); camille.jubelin@etu.univ-nantes.fr (C.J.); marie.francoise-heymann@ico.unicancer.fr (M.-F.H.); 3European Associated laboratory “Sarcoma Research Unit”, INSERM, Department of Oncology and Metabolism, Medical School, University of Sheffield, Beech Hill road, S10 2RX Sheffield, UK

**Keywords:** circulating tumor cells, spheroids, organoids, preclinical models, tumor heterogeneity, personalized medicine

## Abstract

The main cause of death from cancer is associated with the development of metastases, resulting from the inability of current therapies to cure patients at metastatic stages. Generating preclinical models to better characterize the evolution of the disease is thus of utmost importance, in order to implement effective new cancer biomarkers and therapies. Circulating Tumor Cells (CTCs) are good candidates for generating preclinical models, making it possible to follow up the spatial and temporal heterogeneity of tumor tissues. This method is a non-invasive liquid biopsy that can be obtained at any stage of the disease. It partially summarizes the molecular heterogeneity of the corresponding tumors at a given time. Here, we discuss the CTC-derived models that have been generated so far, from simplified 2D cultures to the most complex CTC-derived explants (CDX models). We highlight the challenges and strengths of these preclinical tools, as well as some of the recent studies published using these models.

## 1. Introduction

Circulating Tumor Cells (CTCs) are cancer cells that have escaped from a primary tumor or metastatic site. Some of them can survive in the bloodstream, migrate into the interstitial space (extravasation process) and finally result in the formation of a distant tumor in a new micro-environment [1]. Based on this context, isolating and characterizing CTCs at the molecular/functional level may be the key for future therapeutic developments in oncology [2]. The molecular characteristics of CTCs evolve as the tumor foci progress and throughout tumor progression. They express new sets of clusters of differentiation and mutation profiles which are related to the emergence of minor new sub-clones that fuel tumor heterogeneity. Consequently, CTCs partially reflect the spectrum of tumor mutations and its heterogeneity, but can be considered as a snapshot of the evolution disease at a given time [3]. CTCs could thus be genotyped and functionally characterized to study and target the evolving mutational landscape of primary and/or metastatic tumors [4]. In the past decade, numerous clinical trials have demonstrated the clinical/biological value of CTCs enumeration. Indeed, even CTC counting is not a common practice in oncology, CTCs may be very informative as biomarkers in the follow-up of malignancies [5,6,7,8]. In addition, CTCs can easily be collected at any stage of the disease by means of a non-invasive liquid biopsy. Overall, the studies revealed the promising potential utility of CTCs to adjust treatment depending on their molecular profile [5,9]. All these characteristics make CTCs very attractive for generating in vitro and in vivo models for studying different areas of cancer research, such as therapy, disease evolution or real time genomic characterization.

Although CTCs have been identified and studied in most malignancies, there is still a lack of firm knowledge concerning the biological characteristics of these cells and their life cycle. In particular, there is uncertainty regarding the time point of their first release into the bloodstream, their genetic profile in relation to the bulk tumor, the putative modes of intravasation and extravasation, and their means of survival in circulation. Their low frequency in blood, heterogeneity, and poor survival, as well as the challenging methods for isolating them, make them difficult to characterize exhaustively in transcriptomic, genomic and functional terms. In this context, improved methods for CTC culture and expansion are mandatory to investigate their molecular profile and characterize the control of their behavior by the role of the local microenvironment. Despite these limitations, different in vitro and in vivo models of CTCs have been developed in the last decade [10,11]. In the present review, we will focus on the status of the methodologies for CTC enrichment and isolation, and we will describe the most commonly-used methods for establishing CTC-derived models, as well as their main advantages and disadvantages. Future perspectives will also be discussed.

## 2. Current Methodologies for CTC Enrichment and Isolation: Pros, Cons and Improvements Needed

CTCs are extremely rare populations present in the blood of cancer patients. The existence of one CTC in a background of billions of blood cells has been described [12]. One of the main technical challenges, one that has still not been fully resolved, involves the successful enrichment and isolation of CTCs. However, in the last few years there have been some improvements in the development of these methodologies, which are described extensively in numerous published reviews [13,14,15,16]. Methods for CTC capture are based on differences in biophysical or biological properties between CTCs and normal blood cells. However the high grade of heterogeneity in CTCs has challenged the utility of these technologies for isolating pure and representative CTC subpopulations [17]. Here we will emphasize the main advantages and pitfalls of these technologies, as well as recent improvements.

### 2.1. Biophysical Property-Dependent Enrichment Methodologies

Biophysical property-dependent enrichment methodologies rely on the ability to discriminate between CTCs and other cells based on physical characteristics such as density, size, deformability, and electric charge. The larger size and stiffness of CTCs in contrast to leukocytes have been exploited in past decades to develop microfiltration-based devices in two and three dimensions. In these methods, blood is filtered through pores that trap molecules larger than the maximum pore sizes. ISET^®^ (Paris, France) [18,19], ScreenCell^®^ (Sarcelles, France) [20,21], CellSieve™ (Rockville, MD, USA) [22,23], Flexible Micro Spring Array (FMSA) [24], Parsortix™ (Angle PLC, Guildford, UK) [25,26], Resettable Cell Trap [27] and Cluster Chip [28] are some of the devices available on the market. Filtration does not capture CTCs with a size the same as, or smaller than, the pore diameter. Moreover, the fact that some leukocytes have a similar density and size as CTCs may reduce the purity. In addition, hemodynamic stress can damage CTCs, reducing the viability rate. Processing large volumes may cause the membranes to clog. Nevertheless, filter-based methods have some important advantages such as ease-of-use, low cost, fast processing, high-throughput, and good recovery efficacy of CTC clusters, as well as being a very convenient technique for users. The most recent improvements include 3D microfilters, which reduce the hemodynamic stress on cells, thus sustaining cell viability, and the implementation of CTC size-amplification strategies, which reduce the loss of small-sized CTCs [29].

Density gradient centrifugation enriches CTCs in the mononuclear fraction, taking advantage of their similar buoyant density. Ficoll-Paque^®^ (GE Healthcare Life Sciences, Freiburg, Germany), Oncoquick^®^ (Greiner Bio One, Courtaboeuf, France) [30], RosetteSep™ (STEMCELL Technologies, Grenoble France) [31,32], AccuCyte enrichment and CyteSealer™ (RARECYTE, Seattle, WA, USA) [33,34] are the most representative examples. The reliability, ease-of-use, inexpensiveness and suitability for high throughput of gradient centrifugation have made these widely-used methods. However, the main limitations of these methods are the loss of high-density CTCs, and the inefficiency in eliminating leukocyte contamination, resulting in low purity. As a result, these methods are commonly used as an initial step before further isolation methods. Some upgrades in centrifugation methodologies include depleting leukocytes by means of specific antibodies to improve purity, and inserting a porous barrier into the centrifuge tube to reduce the loss of larger CTCs or clusters [30,31,32,33,34,35,36].

Di-electrophoresis [DEP] cell separation technology has been used to enrich and isolate CTCs by using dissimilarities in morphology and electrical properties in the different cell types. The ApoStream^®^ (APOCELL, Houston, TX, USA) [37,38] and Deparray™ (Menarini-Silicon Biosystem, Bologna, Italy) [39,40] systems are the most used. The most remarkable feature of these methods is the high viability of the cells isolated, and the possibility of isolating single cells. The drawbacks include the electrical properties of CTCs and normal blood cells, which partially overlap giving low purity rates compared to label-dependent methods. They also require cumbersome sample preparation procedures. Emerging methods based on DEP have been simplified, with cost-efficient manufacturing devices and better cell capture performances [41].

### 2.2. Biological Property-Dependent Enrichment Methodologies

Biological property-dependent enrichment methodologies can be a positive or negative selection procedure. These technologies are based on immunomagnetic or microfluidic devices where the specific antibody is attached. Positive enrichment methodologies are based on targeting surface markers that are only expressed on CTCs. Many affinity-based enrichment technologies use epithelial markers which may be down-regulated during epithelial-mesenchymal transition (EMT). To overcome this, the most recent technologies include other markers, such as stem cell markers, mesenchymal markers or cancer specific antigens. CellSearch^®^ (Menarini-Silicon Biosystem, Bologna, Italy) [42,43], the only device approved by the FDA, AdnaTest [44], MagSweeper [45,46], CTC-Chip [47], GEDI [48,49], OncoCEE™ (Biocept, San Diego, CA, USA) [50], Herringbone Chip [51,52], Ephesia [53], Magnetic Sifter [54,55], IsoFlux [56,57], CTC-iChip [58,59] and Gilupi CellCollector™ (GILUPI GmbH, Postdam, Germany) [60,61] are some of the positive selection methods currently available. The major advantage is the high purity obtained. However, they are not able to isolate the entire population of CTCs due to high heterogeneity, mainly related to the EMT process and secondary genetic/epigenetic events [2]. In addition, the epitopes are frequently inaccessible in clusters of CTCs and consequently cell clusters may be less frequently detected.

Negative selection technologies overcome the assumption regarding the unknown nature of CTCs, as they are based on the depletion of blood cells (mainly leukocytes) that are better characterized, using antibodies against antigens expressed in these cells. The EasySep™ depletion kit (STEMCELL Technologies, Grenoble France) [62,63], QMS [64] and CTC-iChip [58,59] are widely-used negative enrichment methods. This method has the potential to enrich all CTC subpopulations, but on the other hand, it results in low purity.

## 3. In Vitro CTC-Derived Models

Zhang et al. were the first to establish primary cultures from CTCs obtained from patients with advanced stage breast cancer [65]. They isolated, established long-term cultures of human breast cancer CTCs, and identified markers for the brain metastasis signature. Cultured CTCs generated brain and lung metastases when they were injected either intracardiacally or into the tail vein of immunodeficient mice. The results suggested the potential use of this signature to predict which circulating cells have the ability to metastasize [65]. Similarly, in another study performed by Cayrefourcq et al., a stable CTC line derived from a colon cancer patient was established [11]. This CTC line shared important common features with the tumor cells analyzed ex vivo, and was able to induce in vitro angiogenesis and tumors in vivo [11]. Other studies have demonstrated the successful establishment of primary cultures of CTCs derived from esophageal cancer [66], small cell lung cancer (SCLC) [67], urinary bladder cancer [68], pancreatic cancer [69], gastric cancer [70] or malignant pleural mesothelioma [71] patients.

Even the genetic profile of CTCs can be different from their primary tumor counterparts, CTCs are representative of the current state of disease [3]. CTCs closely mimic the genetic features of tumors at a given time, making it possible to perform functional investigations. In general, 2D cultures are simple and low-cost. Some of the problems associated with establishing primary cell lines are difficult isolation and short life span. An exceptional case is SCLC, which is distinguished by the release of excessive amounts of CTCs in advanced stages. This has made it possible to establish several permanent cell lines in vitro [72]. Moreover, in 2D cultures cell-cell and cell-extracellular environment interactions are not represented as they would be in the tumor mass, which is the main limitation as the importance of these interactions has been demonstrated in cell differentiation, proliferation, expression of genes and proteins, responsiveness to stimuli, drug metabolism and other cellular functions [73]. Once cells are attached to the plastic surface, their morphology is completely altered, which can affect the organization of the structures inside the cell and may impair their functions [74]. In addition, in 2D cultures the spatial heterogeneity is lost. The availability of oxygen, nutrients, metabolites and signal molecules in 2D cultures is unlimited, in contrast to cancer cells in vivo where there is more variable availability of nutrients due to the architecture of the tumor mass, leading to modifications in the molecular expression patterns of cancer cells, as well as altered behavior. In addition, the 2D environment makes it complicated to establish clinically-relevant cell lines [75] capable of surviving several cell divisions in adherent monolayer culture CTCs [76].

To bypass the technical constraints associated with 2D cultures, Zhang et al. generated a three-dimensional [3D] co-culture model based on early lung cancer CTCs, tumor-associated fibroblasts and extracellular matrix proteins [collagen I and Matrigel] to establish a tumor micro-environment that facilitated CTC expansion and better mimicked in vivo tumor growth [77]. The authors found concordance in the mutations of the key genes involved in lung cancer progression, such as cytokerain-8, cytokeratin-18, thyroid transcription factor 1 (TTF-1) and epidermal growth factor receptor (EGFR), both in expanded CTCs and primary tumors. This result validated the advantages of 3D culture methods when studying metastatic disease progression.

### 3.1. Spheroids or Tumorospheres

Spheroids or tumorospheres are a type of 3D cell modeling constructed from tumor cells alone or in combination with other cell types with or without scaffolds. They simulate live cell environmental conditions, as they are based on the creation of spheroid structures in which cell-cell and cell-environment interactions are maintained, as well as cell polarity and morphology [78] (Figure 1). 

Several methods have been developed to generate spheroids. They can be divided into two main groups: scaffold-based spheroids, such as matrix on top, matrix embedded, matrix-encapsulation, spinner flasks and micropatterned plates; and scaffold-free spheroids, such as ultralow attachment plates, hanging drop, magnetic levitation, and magnetic 3D printing [78]. Multiple studies have demonstrated the feasibility of the culture of CTC-derived spheroids [72,76,79,80]. Klameth et al. established 5 CTC cell lines from patients with recurrent small cell lung cancer (SCLC) that developed tumorospheres spontaneously and showed the typical markers associated with SCLC, such as synaptophysin, enolase-2 and chromogranin A, as well as high resistance to the chemotherapeutics commonly used in the treatment of SCLC. The authors concluded that CTC-derived tumorospheres provided in vitro equivalents of actual in vivo multicellular structures and could thus be used to study metastases via CTCs, drug resistance and advanced therapeutic modalities for attacking 3D-tumors [72].

Similarly, Yu et al. isolated and cultured CTC-derived tumorospheres from different breast cancer patients and performed drug sensitivity assays. They observed different patterns of drug susceptibility linked to the genetic context of each patient [73]. In another study, Zhang et al. developed a clinically-validated method for isolating hepatocellular carcinoma (HCC) CTCs using a microfluidic device. They demonstrated the feasibility of using isolated CTCs to grow spheroid-like structures and assess chemotherapeutic agents for HCC treatment [79]. Vishnoi et al. reported the existence of different CTC subsets with distinct capacities for generating tumorospheres and long-term in vitro growth, depending on uPAR/int β1 expression [80]. In addition, molecular characterization of these CTC subsets resulted in differences in cell adhesion and invasion ability, relevant to breast cancer brain metastasis disease. Overall, these studies have reported similarities in the morphology, gene expression, cell signaling, metabolism and behavior of spheroids and cells growing into a tumor mass [81,82]. Moreover, spheroids present cellular heterogeneity and make simple and inexpensive biological research possible. Despite their advantages, they nevertheless have certain disadvantages, such as: poor homogeneity in spheroid size; challenges in drug distribution along the spheroid; low efficiency and repeatability; short life-span, and less work practicality compared to 2D culture systems because they require special care in handling [78,83] (Table 1).

### 3.2. Organoids

Organoids are another 3D type culture model. Organoids were originally defined as 3D structures formed from cells isolated from a piece of fresh tissue, or from embryonic or pluripotent stem cells. Such cells should self-organize into three-dimensional conditions to facilitate their self-renewal and maintain their differentiation properties [84] (Figure 1). Consequently, organoids can partially reproduce the organization of the organ with structures (e.g., glands) differentiated from those from which they originate. Organoids were developed successfully from CTCs by Gao et al. [85]. They successfully established seven organoid lines of prostate cancer from biopsy specimens and circulating tumor cells that summarized the molecular heterogeneity of prostate cancer subtypes, useful for performing genetic and pharmacological studies. Interestingly, they found a high frequency of RB and TP53 pathway dysfunctions in the organoid lines, suggesting that drugs targeting these pathways should become a therapeutic priority [85]. CTC-derived organoid cultures are established in a relatively short time frame, are easy to manipulate, propagate and store, are biologically stable, are suitable for high-throughput screening assays [86] and can be genetically manipulated. They are thus useful for establishing cancer models [87] for potential future applications such as identifying the “driver mutations” involved in cancer development [88]; for modeling metastatic progression and drug-induced selection in patients, by establishing multiple organoid lines from the same patient over a period of time [93]; for studying genetic instability [94]; for performing the screening of specific drugs in cancer patients based on genetic profiles to monitor the evolving mutational landscape and drug sensitivity patterns and thus customize therapies for individual patients [85,89,94,95]. However, CTC-derived organoid cultures still have some inherent limitations. They lack the complexity of the in vivo immune system, and vascularization and high-throughput screening is difficult (Table 1). Further investigations are needed to establish co-culture organoid models with immune or cancer-associated cells, as a more precise model for translational research and drug discovery. In addition, they do not represent the complete spatial heterogeneity of the tumor.

### 3.3. In Vivo CTC Derived Models

Patient-derived xenograft (PDX) models are freshly resected primary tumor or metastasis fragments which are subcutaneously or orthotopically implanted into immunocompromised mice [90]. The most recent methods for tissue biopsies (e.g., needle biopsies) restrict their use to diagnosis, and limit access to them for research investigations. Similarly, access to metastatic materials is frequently limited or impossible because of the invasiveness of the tissue sampling. Based on the concept previously described for spheroids/organoids, CTC-derived explant (CDX) models have emerged recently, in which CTCs are enriched from the blood of patients and injected into immunocompromised mice to generate tumors and expand the initial material [95] (Figure 1). CDX are considered to be models for metastasis, because in order to generate them, the cells used have already overcome part of the selection process for metastatic progression (CTCs). 

One of the main advantages of CDXs is that they can be derived from CTCs collected at different time points during patient follow-up, making it possible to generate paired models that summarize the evolution of the patient’s tumor. This benefit can be exploited for personalized preclinical research, as well as for testing drug efficacy, and developing predictive biomarkers for standard and innovative anticancer drug-based therapies [91]. Additionally, CTCs are the reflection of the overall heterogeneity of the tumor in contrast to a small biopsy. Heitzer et al. compared the genetic profile of CTCs, primary tumors and metastases and concluded that most mutations detected in CTCs were also present at subclonal level in the primary tumors and metastases from the same patient [92]. However they also present certain limitations, the most important being the delay that exists between patient treatment and tumor engraftment; the generation of lymphomagenesis of human tumors in mice; and the fact that it is a high cost and time-consuming model and thus not suitable for high-throughput drug screening [96]. In addition, the clinical validity of CTC-derived PDX is limited by the fact that CTCs in circulation are subjected to an active immune escape program which is not reflected when they are injected into highly immunodeficient mice, and may result in different disease progression [97]. Similarly, the influence of micro-environmental conditions limits the value of CDX models, as CTCs are applied subcutaneously and grow in a murine milieu of growth factors and conditions in the absence of interaction with cells present in the initial tumor micro-environment [98]. Furthermore, generating xenografts seems to require high amounts of CTCs, which is a limitation for the vast majority of cancers for which only a few CTCs are detectable. This explains why such models are sometimes only possible for patients with advanced disease. However, despite these technical limitations, several studies have proven the clinical value of CDX models (Table 1). Hodgkinson et al. enriched CTCs from SCLC and developed specific CDXs. Tumor-bearing mice can be considered as patient avatars. The tumors developed in mice showed responses to platinum and etoposide that were similar to those of the patients from which they originated [95]. In a similar study, Lallo et al. generated CDX models to test a new combination of drugs for the treatment of SCLC. This resulted in successful treatment for some of the models, and it could be used as an excellent alternative for patients with poor responses to standard chemotherapy [99]. A very recent published work by Drapkin et al. shows efficient generation of 34 CDX models of SCLC that accurately summarized both the genomic and functional features of patient tumors. In addition, serial models derived from an individual patient at multiple time points reflect the evolving clinical response of that patient’s tumors [100]. Importantly, CDXs from individual patients shared genomic alterations, but displayed the intratumoral and especially intertumoral heterogeneity found in patients. Such heterogeneity is clinically relevant and must be taken into consideration, given its impact on treatment, chemoresistance, dissemination and metastasis formation [101,102].

## 4. Discussion and Future Perspectives

Patient-derived CTC cultures were first established in 2013 [65]. Despite the initial promise, implementing this procedure into clinical practice has been challenging due to the low efficiency of these methods and the prolonged periods required for cell line establishment. Currently one of the main challenges for developing CTCs derived in vitro, and in vivo models, is the development of innovative tools for the quick isolation and characterization of the CTCs. Many technologies have been developed in recent years, and their advantages and disadvantages are briefly described in this review. Interestingly, Shen et al. recently published a study comparing different methodologies for CTC isolation in terms of the efficiency of CTC recovery, purity, and the CTC concentration limits that could be detected in blood [14]. It is important to note that due to the high diversity of CTCs, either in phenotype or genotype, there is no perfect method. The main criteria for selecting the enrichment/isolation method is the high preservation of the CTCs, which represents a very fragile population, and the speed of the isolation process, as the faster cells are isolated, the more viability they have [36,103,104,105]. To increase the preservation of CTCs, Streck^TM^ (STRECK, Omaha, NE, USA), and CellSave^TM^ (Menarini-Silicon Biosystem, Bologna, Italy) blood collection tubes have been designed [106]. To reduce the duration of the process, Lin et al. described a portable filter-based microdevice for fast detection and characterization of CTCs [107].

The high heterogeneity of CTCs must be taken into account before selecting the isolation and enrichment method/s in order to minimize biases in this early step—A step that has a high impact on the final results (e.g., yield, purity). It is also important to consider that despite the efforts made in the development of new methodologies, most of them are initially validated in spiked cells from cell cultures, thus they will always overpredict the device’s performance, as cancer cell lines tend to be more homogenous and more physically different from white blood cells than patient CTCs [15]. Other important challenges are maintaining viability and keeping intact the cell surfaces of isolated CTCs. In this regard, several authors have used the filtration-based MetaCell^®^ device (Ostrava, Czech Republic) to isolate and culture CTCs from patients with prostate cancer [108], gastric cancer [70], lung cancer [109], bladder cancer [68] or gynecological cancers [110], without needing to detach cells from the filter, and thus keeping the cell surface unaffected.

The most cutting-edge development for efficient expansion of patient CTCs in vitro is based on a microfluidic system, where autologous immune cells and cancer cells are co-cultured making the establishment rapid and efficient [111]. The main advantage of this method is that it does not require prior enrichment of CTCs from blood samples. In addition, the simplicity and speed of cluster formation makes this method feasible for routine use in clinics for evaluating anticancer treatment. However, the authors described some limitations to its applicability, as the potential for forming clusters varied within the clinical CTC samples due to biological heterogeneity and disease conditions (e.g., patients with more than 4 lymph nodes provide samples that have a higher potential for forming clusters) [112].

Wang et al. raised the issue of culturing isolated CTCs for clinical purposes solely from peripheral blood, as the structure of the circulation system entails that not all CTC populations are uniformly distributed in blood, and using peripheral CTCs alone would not represent the entire genetic variability of this population [113]. This has been supported by Sun et al. who revealed, in a recent publication, the high spatial heterogeneity in the epithelial and mesenchymal composition of CTCs from different vascular sites, which results in distinct clinical significance in HCC [114]. For more accurate clinical use of CTCs, future studies should focus on analyzing CTCs from different vascular sites.

Before establishing CTC-derived cultures, it is important to consider the patient’s treatment status, as the failure of CTC cell line generation during the early stages of treatment, caused by the reduced tumor burden, has been demonstrated. In addition, it is necessary to optimize CTC culture conditions for a suitable proliferation index and low cell mortality [76]. Importantly, in the study carried out by Min et al., the authors tested different culture conditions and concluded that CTCs best as tumorospheres when they were cultured under hypoxia with serum-free media supplemented with epidermal growth factor and basic fibroblast growth factor. In addition, they observed the senescence of the cells after a few cell divisions when they were cultured in an adherent monolayer [76]. These observations could be explained by the fact that CTCs cultured as non-adherent spheres may better reflect the intrinsic properties of tumor cells, which remain viable in the bloodstream after losing their attachment to the basement membrane. Spheroids exhibit increased resistance to chemotherapeutics and irradiation due to the presence of quiescent cells, cell contact-mediated effects and the lack of generation of oxygen radicals. CTCs grown in vitro as spheroids could be used to analyze the chemoresistance phenotype of native solid tumors. Indeed, it has been shown that spheroids display pathways of resistance linked to hypoxia, altered chromatin structure, impairment of apoptosis, cell cycle alterations and decreased drug perfusion [115,116].

For a more realistic approach to in vitro and in vivo models using CTCs, the tumor micro-environment should be considered. Lovitt et al. described a standardized and highly reproducible extracellular matrix-based 3D model that better recreates the in vivo micro-environment and tumor biology compared to monolayer cell cultures [117]. Another emerging method that makes it possible to capture tumor micro-environment heterogeneity is “3D bioprinting”. This method is based on the generation of 2D patterns containing cells and other bioactive factors, which are stacked to form complex 3D structures mimicking heterogeneous tissue structures that summarize the features of the micro-environment [118]. Further efforts to understand the impact of the micro-environment on tumor progression may make it possible to generate predictive data from more biologically-relevant models that incorporate the multicellular constituents and physical properties of a tumor.

CDXs have recently emerged in in vivo models. One interesting application for CDXs is the generation of new models of drug resistance, as shown by TerBrugge et al. in PDX models [119]. The comparison between genomic analysis from resistant models and relapsed patients can be useful for unraveling clues to tumor progression and cancer resistance mechanisms. However, as CDX models are established in immunodeficient mice they present limitations for therapy testing. This may be solved by using the humanized mouse models that are starting to enter the research field. These mice are normal, immunocompromised mice into which a human immune system has been engrafted [120]. Generating CDXs may improve the possibility of predicting the response to specific therapies. In vitro and in vivo models derived from CTCs are promising approaches to analyze the tumor heterogeneity, to predict the therapeutic responses however clinical trials are mandatory to confirm their clinical potential and to allow their use in clinical practices.

## Figures and Tables

**Figure 1 cancers-11-00019-f001:**
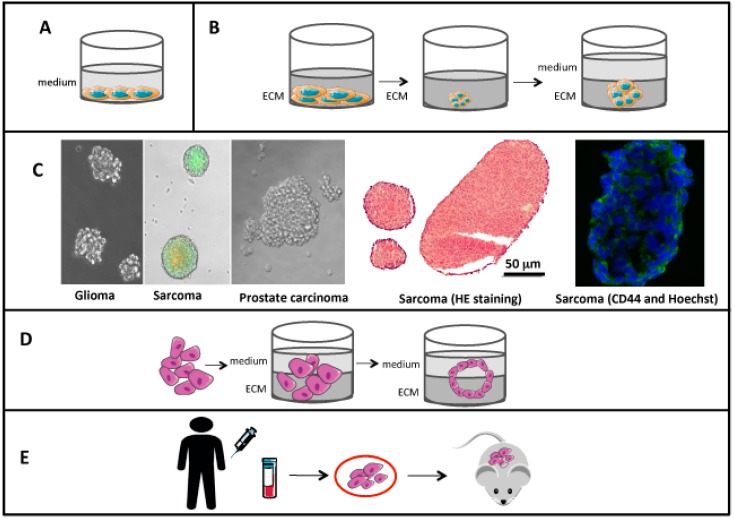
CTC-derived pre-clinical models. (**A**) 2D cultures. (**B**) Spheroid generation. Cell suspension is cultured in extracellular matrix (ECM) or in a specific liquid culture medium. After a couple of days, cells assemble into spheroids. Specific growth factor–supplemented medium is added to allow the spheroids to expand over time. (**C**) Images of spheroids derived from isolated CTCs of different tumor origin: U251 glioma, KHOS sarcoma and PC3 prostate carcinoma cell lines (original magnification: X200). Histological image of KHOS spheroids (HE stained) and CD44 (green) expression by a KHOS spheroid observed using confocal microscopy (blue: Hoechst staining of nuclei). (**D**) Organoid generation. Suspensions of isolated cancer cells are cultured in the presence of specific growth factor–supplemented medium and ECM. After 7-10 days, the generation of cancer-organoid structures can be observed. (**E**) CDX models. CTCs are collected from the patient by means of a non-invasive biopsy. They are isolated and injected into immunodeficient mice that may form a tumor.

**Table 1 cancers-11-00019-t001:** In vitro and in vivo CTC-derived models. Advantages and disadvantages.

Model	Advantages	Disadvantages	References
**CTC-2D cultures**	- Mimic the genetic features of the initial tumors - Simple and low-cost	- Short life span - Do not represent cell-cell and cell-extracellular environment interactions - Alteration to cell morphology due to the adherence step on the plastic surface - Loss of cell heterogeneity - Unlimited availability of oxygen and nutrients; - Do not respect spatial heterogeneity	[70,71,72,73]
**Spheroids or tumorospheres**	- Maintained morphology, gene expression, cell signaling and behavior compared to cancer cells in the tumor mass - Allow High-throughput drug screening - Inexpensive - Can be manipulated genetically	- Low efficiency - Low repeatability (difficulty to reproduce spatial organization) - Short life-span - Incomplete micro-environment	[74,81]
**Organoids**	- Biologically stable - High-throughput drug screening - Can be manipulated genetically	- Lack the complexity of the in vivo immune system and vascularization - May lack key cell types - Not very suitable for high-throughput	[84,85,86,87,88]
**CDX models**	- Mimic tumor evolution - Useful for studying the metastatic process	- Delay in tumor engraftment and patient disease progression - May cause lymphomagenesis - High cost and time-consuming model - Do not allow high-throughput drug screening - High amount of CTCs required - Lack the complexity of the human immune system	[89,90,91,92]

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
