# Peer review of "Circulating Tumor Cell-Derived Pre-Clinical Models for Personalized Medicine"

_cancers, 2018, doi:10.3390/cancers11010019_

Round 1
Reviewer 1 Report
The authors provide a deep revision of the main methods for enrichment, isolation and modeling of CTCs. They clearly state the importance of CTCs in cancer, as CTCs offer a quick and non-invasive opportunity to study tumor heterogeneity and evolution, metastatic risk or treatment efficacy, during disease progression. The manuscript summarize the available techniques with special attention to the weaknesses and strengths of every methodology. The thorough revision of the topic converts the manuscript in an effective guide for every scientist willing to introduce themselves to CTC research.
Although the presence of CTCs have been described for a long time, the study of their implications in preclinical studies have been hampered by the lack of affordable and easy to use isolation methods. The low frequency of CTCs together with modest purity ratios and reduced viability after isolation, are the main challenges that researchers have to face when they join the field of CTCs. Recent advancements have contributed to the development of quicker and more effective tools for CTCs isolation and characterization. Therefore the generation of CTC-derived models for CTC expansion and subsequent preclinical studies, is original and timely. In addition, the information is well written and well presented. The manuscript is pleasant and exhaustive, making it accessible and informative. This review is the kind of text I would recommend to anyone moving into the research of CTCs.
Author Response
Wet thank the reviewer for his/her very positive comments
Reviewer 2 Report
This review provides a through outline regarding the emerging utility of CTC-derived models for personalized patient care, including the advantages and disadvantages of various systems. Additionally, the authors provide a succinct, yet comprehensive, account of the state of technology in the CTC field. Finally, the authors present the reader with important considerations that must be evaluated when selecting a CTC assay and challenges that still exist in utilizing CTC-derived models.
1. Paragraph 1. Line 40.
”…but can be considered as a photo of the evolution in tumor mass at a given time”.
This sentence structure is a bit odd. Would suggest changing it to
“…and can be considered a snapshot (or photo [if preferred]) of the evolution of the tumor(s)/disease at a given time”
2. Paragraph 1. Line 42.
“The considerable significance of CTCs consists in their biological value.”
It’s unclear what the authors are trying to convey with this sentence. It should be reworded for clarity.
3. Paragraph 1. Line 46.
“They can also be used to adjust treatment depending on their molecular profile.”
To my knowledge, the use of CTC molecular profiling to augment patient treatment has not yet been demonstrated in a large randomized clinical intervention study, and therefore this sentence should be reworded to indicate that this is a promising potential utility of CTCs. The way is it stated here implies that it is common practice in the field.
4. Paragraph 2. Line 51.
“…of their cells and their life cycle.”
Should be “… of these cells and their life cycle.”
5. Paragraph 2. Line 56.
“In this context, improved methods for CTC culture and expansion could make possible molecular, cellular and behavioral investigations with high accuracy and reproducibility.”
It’s unclear what the authors are trying to convey with this sentence. It should be reworded for clarity.
6. Paragraph 3. Line 68.
“…there have been huge improvements…”
Although improvements have been made in CTC technologies, I would argue these improvements have not been huge, especially from a clinical perspective where the CellSearch system is still the gold standard technology and the only CTC technology with FDA approval.
7. Paragraph 4. Line 82.
“Filtration does not capture CTCs with a size the same as, or smaller than, the pore diameter. This includes metastatic cells, which become more deformable”.
It is unclear what the authors are referring to when they mention metastatic cells. If they are referring to more mesenchymal cells, this statement is incorrect. The Parsortix system, for example, has been recognized for its ability to capture these cells as it is not reliant on EpCAM for capture. This statement should be reworded or omitted all together. If included, it needs to be properly referenced.
8. Paragraph 4. Line 88.
“There is also permanent development.”
It’s unclear what the authors are trying to convey with this sentence. It should be reworded for clarity.
9. Paragraph 5. Line 92.
“…enriches CTCs in the monocyte fraction…”
Should be “…enriches CTCs in the mononuclear fraction…”
10. Paragraph 5. Line 93.
“…accuyte enrichment…”
Should be “…Accucyte enrichment…”
11. Paragraph 5. Line 94.
“… the most representative devices…”
Should be “… the most representative examples…”, not all of these are “devices”
12. Paragraph 5. Line 98.
“Some upgrades in centrifugation…”
References should be included at the end of this statement.
13. Paragraph 7. Line 115.
“…but they are down-regulated during epithelial-mesenchymal transition.”
As written, this sentence implies that epithelial markers are always down-regulated on all CTCs. However, many retain very high epithelial marker expression. Therefore this sentence should be changed to “…which may be down-regulated during epithelial-mesenchymal transition.”
14. Paragraph 7. Line 124.
“…CTCs that form clusters are lost…”
This statement is untrue. CTC clusters can be detected with these technologies, although they may be less frequent.
15. Paragraph 10. Line 143.
“…that they closely mimic the genetic features of tumors…”
Need a reference for this. Also the authors should clarify that CTCs are more representative of the current state of disease and not all tumors. Their genetic profile can be quite different from their primary tumor counterparts.
16. Line 167.
There is an extra “o” in the word tumorspheres in subtitle 3.1.
17. Figure 1
This figure needs to be higher resolution. The text is difficult to read. Also, in the figure legend, “…that will develop the disease”, should be changed to “…may form a tumor”.
18. Table 1
It is difficult to see the divisions between each model in the disadvantages column. Also, the authors have listed high-throughput as an advantage and a disadvantage of the organoid model.
19. Paragraph 15. Line 242.
“…and limit access to them to research investigations.”
Should be changed to “…and limit access to them for research investigations.”
20. Paragraph 15. Line 244.
“…CTC-derived explant models has emerged…”
Should be changed to “…CTC-derived explant models have emerged…”
21. Paragraph 16. Line 254.
“The existence of mutations has been shown to be found only in the CTC compartment.”
It’s unclear what the authors are trying to convey with this sentence. It should be reworded for clarity.
22. Paragraph 17. Line 294.
“…there is no a perfect method.”
Should be changed to “there is not a perfect method” or “there is no perfect method”.
23. Paragraph 17. Line 294.
“…which represents a very fragile population…” and “…as the faster cells are isolated, the more viability they have…”
References should be included here.
24. Paragraph 17. Line 298.
“In addition, Lin et al. described…”
This statement seems out of place here.
25. Paragraph 21. Line 333.
“…grew best as tumor spheres…”
Should be changed to “…grew best as tumorspheres…”
26. Paragraph 21. Line 341.
”CTCs grown in vitro as spheroids thus perfectly represent the chemoresistance phenotype…”This statement is misleading. Spheroids do not perfectly represent chemoresistance.
27. Paragraph 23. Line 362.
“In vitro and in vivo models derived from…”
The concluding statement in this review is unclear and should be reworded for clarity.
Author Response
We thank the reviewer for his/her very positive comments. Please find below the responses to each comments:
1. Paragraph 1. Line 40.
”…but can be considered as a photo of the evolution in tumor mass at a given time”.
This sentence structure is a bit odd. Would suggest changing it to…
“…and can be considered a snapshot (or photo [if preferred]) of the evolution of the tumor(s)/disease at a given time”
Answer: we modified the sentence accordingly.
2. Paragraph 1. Line 42.
“The considerable significance of CTCs consists in their biological value.”
It’s unclear what the authors are trying to convey with this sentence. It should be reworded for clarity.
Answer: the corresponding sentence has been reworded as follow:
« In the past decade, numerous clinical trials have demonstrated the clinical/biological value of CTCs enumeration. Indeed, even CTC counting is not a common practice in oncology, CTCs may be very informative as biomarkers in the follow-up of malignancies [5-8]. »
3. Paragraph 1. Line 46.
“They can also be used to adjust treatment depending on their molecular profile.”
To my knowledge, the use of CTC molecular profiling to augment patient treatment has not yet been demonstrated in a large randomized clinical intervention study, and therefore this sentence should be reworded to indicate that this is a promising potential utility of CTCs. The way is it stated here implies that it is common practice in the field.
Answer: we do agree with the reviewer and moderated our purposes.
« Overall, the studies revealed the promising potential utility of CTCs to adjust treatment depending on their molecular profile »
4. Paragraph 2. Line 51.
“…of their cells and their life cycle.”
Should be “… of these cells and their life cycle.”
Answer: sentence corrected
5. Paragraph 2. Line 56.
“In this context, improved methods for CTC culture and expansion could make possible molecular, cellular and behavioral investigations with high accuracy and reproducibility.”
It’s unclear what the authors are trying to convey with this sentence. It should be reworded for clarity.
Answer : the sentence has been modified
« In this context, improved methods for CTC culture and expansion are mandatory to investigate their molecular profile and characterize the control of their behavior by the role of the local microenvironment. »
6. Paragraph 3. Line 68.
“…there have been huge improvements…”
Although improvements have been made in CTC technologies, I would argue these improvements have not been huge, especially from a clinical perspective where the CellSearch system is still the gold standard technology and the only CTC technology with FDA approval.
Answer: we do agree and modified the sentence
7. Paragraph 4. Line 82.
“Filtration does not capture CTCs with a size the same as, or smaller than, the pore diameter. This includes metastatic cells, which become more deformable”.
It is unclear what the authors are referring to when they mention metastatic cells. If they are referring to more mesenchymal cells, this statement is incorrect. The Parsortix system, for example, has been recognized for its ability to capture these cells as it is not reliant on EpCAM for capture. This statement should be reworded or omitted all together. If included, it needs to be properly referenced.
Answer: we ado agree with the reviewer, and the second part of the sentence has been omitted
8. Paragraph 4. Line 88.
“There is also permanent development.”
It’s unclear what the authors are trying to convey with this sentence. It should be reworded for clarity.
Answer: this sentence has been deleted
9. Paragraph 5. Line 92.
“…enriches CTCs in the monocyte fraction…”
Should be “…enriches CTCs in the mononuclear fraction…”
Answer: the sentence has been corrected
10. Paragraph 5. Line 93.
“…accuyte enrichment…”
Should be “…Accucyte enrichment…”
Answer: the sentence has been corrected.
11. Paragraph 5. Line 94.
“… the most representative devices…”
Should be “… the most representative examples…”, not all of these are “devices”
Answer : we do agree and the sentence has been corrected
12. Paragraph 5. Line 98.
“Some upgrades in centrifugation…”
References should be included at the end of this statement.
Answer : references have been added in the revised version of our manuscript
13. Paragraph 7. Line 115.
“…but they are down-regulated during epithelial-mesenchymal transition.”
As written, this sentence implies that epithelial markers are always down-regulated on all CTCs. However, many retain very high epithelial marker expression. Therefore this sentence should be changed to “…which may be down-regulated during epithelial-mesenchymal transition.”
Answer: we do agree and the sentence has been corrected.
14. Paragraph 7. Line 124.
“…CTCs that form clusters are lost…”
This statement is untrue. CTC clusters can be detected with these technologies, although they may be less frequent.
Answer: we do agree and the sentence has been corrected.
15. Paragraph 10. Line 143.
“…that they closely mimic the genetic features of tumors…”
Need a reference for this. Also the authors should clarify that CTCs are more representative of the current state of disease and not all tumors. Their genetic profile can be quite different from their primary tumor counterparts.
Answer: we do agree and the sentence has been corrected.
« Even the genetic profile of CTCs can be different from their primary tumor counterparts, CTCs are representative of the current state of disease [3]. CTCs closely mimic the genetic features of tumors at a given time, making it possible to perform functional investigations. »
16. Line 167.
There is an extra “o” in the word tumorspheres in subtitle 3.1.
Answer: word corrected
17. Figure 1
This figure needs to be higher resolution. The text is difficult to read. Also, in the figure legend, “…that will develop the disease”, should be changed to “…may form a tumor”.
Answer: figure 1 with higher resolution is now proposed and the figure legend modified.
18. Table 1
It is difficult to see the divisions between each model in the disadvantages column. Also, the authors have listed high-throughput as an advantage and a disadvantage of the organoid model.
Answer: The Table 1 is correct. organoids allow high-throughput drug screening in contrast to CDX models as indicated in table 1. In the first case, it is an advantage, in the second one, that is a disadvantage.
19. Paragraph 15. Line 242.
“…and limit access to them to research investigations.”
Should be changed to “…and limit access to them for research investigations.”
Answer: sentence corrected.
20. Paragraph 15. Line 244.
“…CTC-derived explant models has emerged…”
Should be changed to “…CTC-derived explant models have emerged…”
Answer : sentence corrected.
21. Paragraph 16. Line 254.
“The existence of mutations has been shown to be found only in the CTC compartment.”
It’s unclear what the authors are trying to convey with this sentence. It should be reworded for clarity.
Answer: the corresponding sentence has been modified.
« Additionally, CTCs are the reflection of the overall heterogeneity of the tumor in contrast to a small biopsy. Heitzer et al. compared the genetic profile of CTCs, primary tumors and metastases and concluded that most mutations detected in CTCs were also present at subclonal level in the primary tumors and metastases from the same patient [95]. »
22. Paragraph 17. Line 294.
“…there is no a perfect method.”
Should be changed to “there is not a perfect method” or “there is no perfect method”.
Answer: sentence corrected.
23. Paragraph 17. Line 294.
“…which represents a very fragile population…” and “…as the faster cells are isolated, the more viability they have…”
References should be included here.
Answer: new references have been added.
24. Paragraph 17. Line 298.
“In addition, Lin et al. described…”
This statement seems out of place here.
Answer: the sentence has been modified.
To reduce the duration of the process, Lin et al. described a portable filter-based microdevice for fast detection and characterization of CTCs [104].
25. Paragraph 21. Line 333.
“…grew best as tumor spheres…”
Should be changed to “…grew best as tumorspheres…”
Answer: sentence corrected.
26. Paragraph 21. Line 341.
”CTCs grown in vitro as spheroids thus perfectly represent the chemoresistance phenotype…”
This statement is misleading. Spheroids do not perfectly represent chemoresistance.
Answer: we do agree and the sentence has been rephrased.
« CTCs grown in vitro as spheroids could be used to analyse the chemoresistance phenotype of native solid tumors. Indeed, it has been shown that spheroids display pathways of resistance linked to hypoxia, altered chromatin structure, impairment of apoptosis, cell cycle alterations and decreased drug perfusion [112,113]. »
27. Paragraph 23. Line 362.
“In vitro and in vivo models derived from…”
The concluding statement in this review is unclear and should be reworded for clarity.
Answer: the sentence has been rephrased.
Reviewer 3 Report
I enjoyed reading the review. It is well written and describes an important research subject. I recommend to publish it after minor corrections.
Figure 1 should have better resolution.
I'd like to see two more figures about CTC models with more details.
Author Response
We thank the reviewer for his/her very positive feedback and as recommended a higher resolution of Figure 1 has been proposed in the revised version of our manuscript.